# Combination of Silicate-Based Soil Conditioners with Plant Growth-Promoting Microorganisms to Improve Drought Stress Resilience in Potato

**DOI:** 10.3390/microorganisms12112128

**Published:** 2024-10-24

**Authors:** Abdullah Al Mamun, Günter Neumann, Narges Moradtalab, Aneesh Ahmed, Fahim Nawaz, Timotheus Tenbohlen, Jingyu Feng, Yongbin Zhang, Xiaochan Xie, Li Zhifang, Uwe Ludewig, Klára Bradáčová, Markus Weinmann

**Affiliations:** 1Department of Nutritional Crop Physiology, Institute of Crop Science, University of Hohenheim, 70599 Stuttgart, Germany; mamun31130@gmail.com (A.A.M.); narges.moradtalab@yara.com (N.M.); dinak444@gmail.com (A.A.); u.ludewig@uni-hohenheim.de (U.L.); markus.weinmann@uni-hohenheim.de (M.W.); 2Research School of Biology, Australian National University, Canberra 2901, Australia; fahim.nawaz@anu.edu.au; 3Beijing Key Laboratory of Growth and Developmental Regulation for Protected Vegetable Crops, Department of Vegetable Science, College of Horticulture, China Agricultural University (CAU), Haidian District, Yuanmingyuanxilu 2, Beijing 100193, China; mirror0406@163.com (J.F.); zyb950326@163.com (Y.Z.); xxc940201@163.com (X.X.); zhifangli7@cau.edu.cn (L.Z.); 4Department of Fertilization and Soil Matter Dynamics, Institute of Crop Science, University of Hohenheim, 70599 Stuttgart, Germany; klara.bradacova@uni-hohenheim.de

**Keywords:** arbuscular mycorrhiza, drought stress, plant growth-promoting microorganisms (PGPM), potato, soil conditioners

## Abstract

Due to shallow root systems, potato is a particularly drought-sensitive crop. To counteract these limitations, the application of plant growth-promoting microorganisms (PGPMs) is discussed as a strategy to improve nutrient acquisition and biotic and abiotic stress resilience. However, initial root colonization by PGPMs, in particular, can be affected by stress factors that negatively impact root growth and activity or the survival of PGPMs in the rhizosphere. In this study, perspectives for the use of commercial silicate-based soil conditioners (SCs) supposed to improve soil water retention were investigated. The SC products were based on combinations with lignocellulose polysaccharides (Sanoplant^®^ = SP) or polyacrylate (Geohumus^®^ = GH). It was hypothesized that SC applications would support beneficial plant–inoculant interactions (arbuscular mycorrhiza, AM: *Rhizophagus irregularis* MUCL41833, and *Pseudomonas brassicacearum* 3Re2-7) on a silty loam soil–sand mixture under water-deficit conditions (6–12 weeks at 15–20% substrate water-holding capacity, WHC). Although no significant SC effects on WHC and total plant biomass were detectable, the SC-inoculant combinations increased the proportion of leaf biomass not affected by drought stress symptoms (chlorosis, necrosis) by 66% (SP) and 91% (GH). Accordingly, osmotic adjustment (proline, glycine betaine accumulation) and ROS detoxification (ascorbate peroxidase, total antioxidants) were increased. This was associated with elevated levels of phytohormones involved in stress adaptations (abscisic, jasmonic, salicylic acids, IAA) and reduced ROS (H_2_O_2_) accumulation in the leaf tissue. In contrast to GH, the SP treatments additionally stimulated AM root colonization. Finally, the SP-inoculant combination significantly increased tuber biomass (82%) under well-watered conditions, and a similar trend was observed under drought stress, reaching 81% of the well-watered control. The P status was sufficient for all treatments, and no treatment differences were observed for stress-protective nutrients, such as Zn, Mn, or Si. By contrast, GH treatments had negative effects on tuber biomass, associated with excess accumulation of Mn and Fe in the leaf tissue close to toxicity levels. The findings suggest that inoculation with the PGPMs in combination with SC products (SP) can promote physiological stress adaptations and AM colonization to improve potato tuber yield, independent of effects on soil water retention. However, this does not apply to SC products in general.

## 1. Introduction

Limited water availability is one of the major challenges of agricultural production, not only affecting arid and semiarid regions of the world. The global water demand is projected to increase by 50% by 2030 [1] due to declining precipitation related to climate change. Drought-sensitive crops with shallow root systems such as potatoes (*Solanum tuberosum* L.) are particularly affected in this context. Potato is considered the world’s third most important food crop in terms of yield and consumption [2,3] with drought-induced losses in yield potential by more than 30% predicted until 2069 [4,5]. Among agronomic measures to counteract these limitations, the application of plant growth-promoting microorganisms (PGPMs) is discussed as a strategy to improve nutrient acquisition and biotic/abiotic stress resilience in crops [6]. Drought-protective effects are well documented, particularly for arbuscular mycorrhizal (AM) associations. In water-deficit soils, AM fungal hyphae facilitate the availability of water and nutrients to plants by improving soil water retention through the release of glomalin proteins, which increase aggregate stability and soil water-holding capacity [7,8]. The small diameter of AM hyphae allows them to penetrate and come into contact with soil pores that are inaccessible to plant roots. This increases plant water use efficiency under drought stress conditions [9,10,11]. Recent studies provide evidence that AM also changes the plant–water relations and reduces drought-induced oxidative damage by activating detoxification systems for reactive oxygen species (ROS) [12,13]. Moreover, they also develop associations with other beneficial soil microbes, also known as mycorrhizal helper bacteria (MHB), mediating synergistic effects on growth promotion and stress resistance in host plants [14]. This applies also to certain strains of the bacterial genera *Pseudomonas* and *Bacillus* with well-documented PGPM functions used as plant inoculants [15,16]. Accordingly, applying microbial consortia based on combinations of inoculant strains of different origins with complementary or synergistic properties is discussed as an optimized strategy to cope with variable environmental conditions [17]. However, limited reproducibility of the expected benefits under field conditions remains a major challenge.

The expression of stress-protective effects induced by beneficial soil microorganisms requires a successful rhizosphere establishment of the inoculants, driven by root activity and root exudation of the host plant. Consequently, stress factors affecting root growth and activity during the sensitive establishment phase can strongly counteract successful root colonization and finally affect the efficiency of the expected protective plant–microbial interactions. This scenario is more likely under open field conditions, influenced by variable environmental factors, compared with protected nurseries or greenhouse cultures where the application of microbial inoculants is frequently more successful [18,19]. Thus, the efficiency of inoculant applications may be increased by any measures that protect both the host plant and the inoculants from stress impacts during the establishment phase. Under drought stress, the use of water-absorbing silicate-based soil conditioners (SCs) has been reported as a protective approach, particularly for dry and sandy soils in arid and semiarid climates [20,21], and commercial formulations are also used for golf greens and ornamental plants [22]. The respective soil additives are made from silicate rock powders in combination with synthetic super-absorbents or organic polymers. The formulations are usually incorporated into the soil in the form of granules or powders. Due to their water-absorbing and swelling properties, SCs can retain additional water in dry soils [21,23]. The application of SCs may not only increase the substrate water-holding capacity but also the formation of soil aggregates and aggregate stability, the infiltration rate, water permeability, soil aeration, and ion exchange capacity, but reduce nutrient leaching, surface water runoff, and soil erosion [24,25]. Particularly under water-deficit conditions, these factors can exert positive effects on nutrient uptake, root activity, and finally, plant growth and development [21,24,26].

Hydrogel SCs and similar biochar amendments may also interact synergistically with PGPMs [27,28,29]. In [30,31], the application of hydrogel soil conditioners improved the microenvironment for soil microbial colonization, which was similarly demonstrated for beneficial microbial communities in the wheat rhizosphere exposed to water limitation [32]. These findings suggest that the use of hydrogel soil conditioners may provide options to support plant–microbial interactions with PGPM inoculants. Accordingly, the first reports describe drought-protective synergisms of selected PGPM-soil conditioner combinations [27,33]. However, other studies also discussed detrimental effects, e.g., on AM associations and potential risks due to the release of toxic acrylate monomers at higher soil temperatures [22,34]. Moreover, aspects of biosafety, persistence, and environmental fate are still under debate concerning the widespread agricultural use of polyacrylate superabsorbers as soil amendments [35].

This study was initiated to characterize exemplarily drought-protective effects of two commercial SC products containing synthetic polyacrylate (Geohumus^®^, Frankfurt, Germany) or natural lignocellulose polymers (Sanoplant^®^). It is expected that SC amendments counteract the frequently limited reproducibility of beneficial plant–PGPM interactions reported even for inoculants with proven drought-protective potential [36]. Potato (*Solanum tuberosum* L.) was used as a drought-sensitive host plant [37] and cultivated under controlled water limitation with and without drought-protective PGPM inoculants (*Pseudomonas brassicacearum* 3Re2-7 and *Rhizophagus irregularis* MUCL 41833 [36]). It was hypothesized that under these conditions, the application of SCs would increase the water-holding capacity in the rhizosphere and thereby support the rhizosphere establishment of PGPMs and drought-protective plant–microbial interactions induced by the inoculants, finally leading to improved drought adaptation of the host plant.

## 2. Materials and Methods

This study comprised three pot experiments conducted under semi-controlled conditions over two years (2019 and 2020) at the greenhouse facility Institute of Crop Science, University of Hohenheim, Stuttgart, Germany (48°42′44.6″ N 9°12′30.4″ E) with an average temperature of 22 °C. The minimum night temperature was 16 °C, and the maximum day temperature reached 28 °C with an average relative humidity of 41%. During winter, additional light was provided for eight hours using 400 W high-pressure sodium lamps (SON-T AGRO 400, Philips Lighting GmbH, Hamburg, Germany). The average light intensity was 275 μmol m^−1^ s^−1^.

Potato seedlings were obtained from true potato seeds (F1 hybrid, B 464, Solynta, Wageningen, The Netherlands) pre-cultivated in a nursery bed and subsequently transplanted in a mixture of silty loam soil, pH (CaCl_2_) = 6.2, CAL extractable P = 34.4 mg·kg^−1^, and CAL extractable K = 81 mg·kg^−1^ dry soil [38], collected from a topsoil layer in the botanical garden at the University of Hohenheim. Before pot filling, the soil was sieved with a 5 mm mesh and later mixed with sand to prepare a substrate mixture of 50% soil and 50% fine sand *w*/*w* (Dorsilit Nr. 9, 0.1–0.5 mm, Waldshut-Tiengen, Germany). The substrate water-holding capacity (WHC) was determined by the method of [39] as described by [36].

### 2.1. Soil Conditioners

Two commercial soil conditioner products, viz. Geohumus^®^ (Geohumus International GmbH, Frankfurt, Germany) and Sanoplant^®^ (SANOWAY GmbH, Dornbirn, Austria), were used as soil amendments to improve the water-holding capacity (WHC) of the substrate. Geohumus belongs to the non-soluble polyacrylates and the synthetic polymer type. It comprises 25% cross-linked, partially neutralized polyacrylates and 75% mineral compounds, i.e., volcanic ground rock and minerals. The proposed effects of this product include the efficient use of water and nutrients, a decrease in irrigation frequencies, and enhanced root growth [40]. Sanoplant is a natural soil additive made from silicate rock powder mixed with organic binding materials, supplemented with specific lignocellulose polymers, and later condensed to variable granules with diameters of 5–90 µm. In moist soil, these granules come in contact with water and disperse to a fine powder to fill the soil macropores, transforming them into micropores that quickly take up large volumes of water to improve the WHC, soil structure, and aggregate stability as well as the microbial activity and nutrient availability of the soil [41,42,43].

### 2.2. Potato Nursery Culture and Transplanting

Potato seedlings were grown in nursery trays for six weeks and first kept in a climate chamber for one week with a 25 °C/16 °C day/night temperature regime until seedling germination. The nursery substrate mixture was prepared by mixing 70% (*w*/*w*) silty loam soil, 25% (*w*/*w*) sand, and 5% (*w*/*w*) peat culture substrate, without fertilization. Each compartment of the tray was filled with a 40 g substrate mixture and one potato seed was sown in each hole, which was then filled with a thin layer of peat culture substrate (sieved with 2 mm mesh) to cover the seed. After one week in the climate chamber, the trays were moved into the greenhouse and the seedlings were grown under controlled conditions: Relative humidity was 41%, and light was supported with halogen lamps in the morning from 8 a.m. to 10 a.m. and in the evening from 5 p.m. to 7 p.m. The maximum temperature was 28 °C during the day and 16 °C at night. The seedlings were watered regularly with distilled water and supplemented with 8 mg N plant^−1^ (Ferty Basisdünger, 3% N as NO_3_^−^), in two installments at 10 and 20 days after sowing to meet the nutritional requirements of the seedlings.

Before transplanting into the main pots, the trays containing potato seedlings were watered and homogenous seedlings were selected for each variant. The seedlings were carefully scooped out from each compartment of the nursery trays along with the adhering substrate material to minimize root injury and immediately planted into the prepared pots (see description of pot experiments). After transplanting, the seedlings were initially grown under well-watered conditions at 60% WHC (establishment phase), followed by a drought phase (15–20% WHC) and a recovery phase (60% WHC) for all pots before harvest. Gravimetric adjustment of water losses was performed daily with demineralized water. The length of the phases, as well as fertilization treatments, were individually adjusted to the duration, pot size, and seasonal variations of the greenhouse conditions of the individual experiments conducted within this study.

### 2.3. Preparation and Application of Microbial Inoculants

To ensure an application rate of 2 × 10^7^ CFU pot^−1^ for *Pseudomonas brassicacearum* 3Re2-7, the bacterial culture was propagated in tryptic soy medium (TSM) broth (Sigma-Aldrich, Darmstadt, Germany) before application. For the liquid culture, agar cubes with bacterial colonies were cut and added into an Erlenmeyer flask filled with 100 mL TSM. The flask was kept on a rotational shaker (Multitron, Infors HT, Bottmingen, Switzerland) at 120 rpm and 28 °C for 24 h. Thereafter, the optical density of the bacterial suspension was recorded at 600 nm (NanoDrop 2000C, Thermo Fisher, Waltham, MA, USA) and adjusted with TSM to reach the absorbance corresponding to 10^7^ cfu mL^−1^. For the bacterial strain, 20 mL of suspension pot^−1^ was applied 10 days after transplanting (DAT) by draining the soil surface close to the stem of the plants. *Rhizophagus irregularis* MUCL41833 was provided as a spore preparation in a formulation of alginate beads containing 25 ± 5 spores per bead or 1000 spores per g fresh weight [44]. The application rate was 12.5 g alginate microbeads pot^−1^ for experiments I and III and 62.5 g alginate microbeads pot^−1^ for experiment II, and alginate beads were mixed with soil substrate before planting.

### 2.4. Pot Experiment I

The first pot experiment was designed to investigate the response of potato plants to microbial inoculants viz. *R. irregularis* MUCL 41833 (AM) and *P. brassicacearum* 3Re2-7 and soil conditioners (Geohumus and Sanoplant) during vegetative growth under drought stress conditions. The substrate consisted of a mixture of soil and sand (1.0 kg each) and the pots were watered according to the desired WHC (see Section 2.3). Fertilizer solutions were applied with rates of 200 mg N kg^−1^ substrate (10% nitrate (Ca (NO_3_)_2_) and 90% stabilized ammonium fertilizer, NovaTec^®^ Solub 21, Compo, Münster, Germany), 150 mg K kg^−1^ (K_2_SO_4_), and 50 mg Mg·kg^−1^ substrate (MgSO_4_) on a dry weight basis. Nitrogen was divided into two split applications with 62.5% of the total amount at the beginning of the experiment and 37.5% at 22 days after transplanting. No additional P fertilizer was added in this experiment. Geohumus (10 g kg^−1^) and Sanoplant (4 g kg^−1^) were also mixed with the substrate according to the instructions of the manufacturers, which were then filled in pots. The desired WHC was adjusted gravimetrically, and the experiment was arranged in a completely randomized design with six variants and five replicates viz. (i) well-watered positive control (PC), (ii) drought negative control (NC), (iii) drought with AM, (iv) drought with AM + 3Re2-7, (v) drought with AM + 3Re2-7 + Sanoplant, and (vi) drought with AM + 3Re2-7 + Geohumus. After an establishment phase (24 d) under well-watered conditions (60% WHC), the plants were subjected to drought stress (34 d at 15–20% WHC) followed by an 11 d recovery (60% WHC).

### 2.5. Pot Experiment II

A second pot experiment was carried out to evaluate the potential of the PGPM consortium used in experiment I combined with Geohumus application to improve the tuber yield of potatoes in the generative phase under water-deficit conditions. For this purpose, the experiment involved larger substrate volumes (15 kg, 50% soil–sand mixture) mixed with the soil conditioner (Geohumus, 10 g kg^−1^ substrate). Adapted to the larger soil volume and the longer duration of the experiment, fertilization was performed with 50 mg P kg^−1^ substrate using Ca(H_2_PO_4_)_2_, 150 mg K kg^−1^ substrate using K_2_SO_4_, 100 mg N kg^−1^ using NovaTec^®^ solub 21 stabilized ammonium sulfate fertilizer (Compo Expert, Münster, Germany), and 50 mg Mg·kg^−1^ substrate using MgSO_4_ kg^−1^ on soil dry weight basis. The experiment consisted of five variants with four replications viz. (i) well-watered control without microbial inoculants and without Geohumus, (ii) drought stress control without microbial inoculant and without Geohumus, (iii) drought stress with soil conditioner Geohumus, (iv) drought stress with microbial inoculants *R. irregularis* MUCL 41833 (Catholic University of Leuven, Belgium) + *P. brassicacearum* 3Re2-7 (Sourcon Padena GmbH, Tübingen, Germany), and (v) drought stress with microbial inoculants *R. irregularis* MUCL 41833 + *P. brassicacearum* 3Re2-7 + Geohumus. After transplanting, the seedlings were watered at 60% WHC for five weeks before the initiation of drought stress. Thereafter, the drought stress variants were kept at 15–20% WHC for 12 weeks. The drought phase was followed by a recovery phase at 60% WHC for two weeks until harvest. The well-watered control plants were continuously watered at 60% WHC throughout the experiment.

### 2.6. Pot Experiment III

The third experiment evaluated the effects of the selected PGPM consortium combined with the natural soil conditioner Sanoplant on plant growth and tuber yield of potato plants under well-watered and drought-affected conditions. The pot experiment was established in culture vessels with 10 kg of substrate (5 kg soil + 5 kg sand). Fertilization was performed at 100 mg N kg^−1^ substrate using 10% nitrate N (Ca (NO_3_)_2_) and 90% N provided by a stabilized ammonium fertilizer (NovaTec^®^ Solub 21, Compo, Münster, Germany), 150 mg K kg^−1^ substrate using K_2_SO_4_, and 50 mg Mg·kg^−1^ substrate using MgSO_4_. The Sanoplant soil conditioner (20 g pot^−1^) was mixed with the top 3 kg of substrate. No additional P fertilizer was added in this experiment. The experiment comprised four replications of five variants viz. (i) well-watered without microbial inoculants and without soil conditioner (PC), (ii) well-watered with inoculants *R. irregularis* MUCL 41833 + *P. brassicacearum* 3Re2-7 and without soil conditioner (AM + 3Re2-7), (iii) well-watered with inoculants *R. irregularis* MUCL 41833 + *P. brassicacearum* 3Re2-7+soil conditioner Sanoplant (AM + 3Re2-7 + SP), (iv) drought stress without microbial inoculants and without soil conditioner (NC), and (v) drought stress with inoculants *R. irregularis* MUCL 41833 + *P. brassicacearum* 3Re2-7 + soil conditioner Sanoplant (AM +3 Re2-7 + SP). For drought stress variants, the three growth phases included (i) a three-week establishment phase at 60% WHC, (ii) a six-week drought phase at 15–20% WHC, and (iii) a two-week recovery phase until final harvest at 60% WHC. The well-watered variants were maintained at 60% WHC during the entire experimental period.

The experimental setup for the three pot experiments conducted in this study is summarized in Appendix A.

### 2.7. Plant Analysis

Harvest of plant material, determination of plant biomass and biomass distribution between shoot, roots, and tubers, root length, analysis of mineral nutrients, UHPLC-MS hormonal profiling, and determination of physiological stress indicators in the plant tissue (total antioxidants, activity of ascorbate peroxidase, concentrations of H_2_O_2_, proline, and glycine betaine) were performed as described by [36].

### 2.8. Mycorrhizal Counting

Root staining was performed on the collected root samples to evaluate the degree of root colonization by arbuscular mycorrhizal fungi. Root tissues were carefully washed and preserved in a 70% (*v*/*v*) ethanol solution. Root samples were selected from different sections of the root system to obtain a representative sample. Aliquots of the root samples were placed inside a plastic box and rinsed with distilled water to remove the ethanol and remaining soil particles adhering to the roots. For bleaching, the root samples were boiled with 10% (*w*/*v*) KOH at 100 °C for 45 min and subsequently rinsed with water. The process was repeated until clear roots were obtained. Bleached roots were acidified with 1–3% HCl and then stained in a 5% ink–vinegar solution (5% (*v*/*v*) ink in 5% (*v*/*v*) acetic acid) at 90 °C for 45 min (Royal blue, Pelikan; Hannover, Germany). De-staining was performed with acidified tap water. The gridline intersection method was used to assess the AMF root colonization, where the stained root sample was distributed on an acrylic glass plate with an engraved grid. The proportion of intersections with intra- and extraradical hyphal structures was counted and calculated according to [45,46].

### 2.9. Statistical Analysis

Statistical analyses were performed using the software program SAS 9.4 University Edition (SAS Institute Inc., Cary, NC, USA). Before SAS analysis, data were analyzed in Sigma Plot (Impixon GmbH, Düsseldorf, Germany) for normality and equal variance testing. For all three experiments, a multiple t-test (t-grouping, *p* ≤ 0.05) was employed to compare the means of treatments for significant differences. Following the methodology outlined by [47], an ANOVA was initially conducted to assess whether the variance between groups exceeded the error variance. If significant F-values were obtained, subsequent t-tests were performed for multiple pairwise comparisons. Experiment I involved six variants with means and SE of five replicates. Experiment II comprised five variants with means and SE of four replicates. Experiment III comprised five variants with means and SE of four replicates.

## 3. Results

### 3.1. Drought-Protective Effects on Vegetative Growth

A first experiment was established for comparative evaluation of drought-protective effects induced by the microbial inoculants with and without application of the selected SC products on vegetative growth of potato plants and the water retention potential of the growth substrate and the rhizosphere soil.

Inoculant treatments promoted fresh shoot biomass production detectable after recovery from the drought period, and here, particularly, the biomass fraction of healthy leaves lacked irreversible symptoms of drought damage (necrosis, chlorosis), with an increasing trend in the order Control < AM < AM&3Re2-7 < AM&3Re2-7+Sanoplant < AM&3Re2-7+Geohumus (Figure 1). No comparable treatment differences were recorded for plant dry matter (Appendix A) since the shoot fresh weight was mainly determined by the fraction of undamaged leaves without symptoms of wilting and necrosis.

Similar trends were recorded in five independent experiments conducted with comparable drought stress regimes (Appendix A). No significant treatment differences were detectable for root biomass (Figure 1) or total root length.

Mycorrhizal colonization was generally high (>90%) even in non-inoculated plants. Application of Sanoplant significantly increased the formation of external AM hyphae (Figure 2) with similar trends also for vesicle formation while Geohumus application was ineffective. However, the stimulation of extraradical hyphal growth by Sanoplant did not improve the P status of the plants, characterized by P shoot concentrations between 2.0 and 2.1 mg·g^−1^ DM in the deficiency range. No limitations were recorded for the remaining mineral nutrients, but the Geohumus treatment significantly increased the K and Mn concentrations and contents in the shoot tissue (Appendix A).

To investigate the effects of the applied soil conditioners on the substrate water-holding capacity, WHC determinations were conducted with the unplanted substrate (Table 1A) and with soil samples collected from the rhizosphere of the potato plants at the end of the culture period (Table 1B). Relative gravimetric water contents showed no significant treatment differences for SC applications at 100, 60, and 20% WHC. In the unplanted substrate, SC application tended to increase the gravimetric water content by 7% compared with the untreated control, which was no longer detectable in the rhizosphere soil obtained by shaking adhering soil particles from excavated root systems at the end of the culture period (Table 1).

At the final harvest, large Geohumus-soil aggregates closely stuck to the fine roots (Appendix A) and could not be removed from the rhizosphere soil. No comparable aggregate formation was recorded for the Sanoplant treatment (Appendix A). This may be attributed to differences in the imbibition characteristics, showing the formation of large aggregates in treatments with Geohumus application but not with Sanoplant after 72 h imbibition in water (Appendix A).

### 3.2. Drought-Protective Effects on Tuber Formation

In a second set of experiments, the drought-protective effects of the selected microbial inoculants and SC products were evaluated in larger soil volumes over extended periods to cover the potential impact on tuber formation. Two independent pot experiments were conducted using Geohumus and Sanoplant soil conditioners, respectively.

#### 3.2.1. Geohumus Experiment

##### Plant Growth AM Colonization and Nutritional Status

Drought stress tended to decrease tuber biomass and tuber numbers by 44 and 48%, respectively (Table 2). Inoculation with the AM+3Re2-7 consortium restored tuber biomass but not tuber numbers to the level of well-watered plants. By contrast, Geohumus treatments harmed tuber biomass production with a significant decline of 68% in combination with microbial inoculants compared with the untreated control. No significant differences were recorded for the tuber numbers.

Despite reduced tuber weight, Geohumus had no negative effects on total plant biomass production. The reduction in tuber biomass in the Geohumus treatment was compensated by increased shoot biomass, particularly expressed in combination with the microbial inoculants (Table 3).

Drought stress significantly reduced AM colonization of the host plants but this effect was compensated by inoculation with AM+3Re27. By contrast, Geohumus application had no effects on AM colonization (Figure 3).

Concerning the mineral nutritional status, P shoot concentrations in the deficiency range were recorded for all experimental variants. N and K concentrations were generally low and close to the deficiency thresholds but were increased by Geohumus treatments (Table 4). Total N, P, and K shoot accumulation was significantly increased in the combination of Geohumus with microbial inoculants (Appendix A). The plants did not show micronutrient deficiencies, however, in the Geohumus variants, Mn (Table 4) and Fe accumulated to tissue concentrations of >400 mg·kg^−1^ DM, close to toxicity levels [48].

##### Physiological Stress Indicators

To assess stress-priming effects potentially induced by investigated amendments, a range of physiological drought stress indicators was determined in the leaf tissue. The selected stress indicators comprised ROS accumulation represented by H_2_O_2_ concentrations, indicators for enzymatic (ascorbate peroxidase, APX), and non-enzymatic (total antioxidants) ROS detoxification, as well as osmotic adjustment (accumulation of proline and glycine betaine).

Even after two weeks of drought stress recovery, ROS (H_2_O_2_) concentrations in the leaf tissues were still massively increased (1500%) compared with well-watered plants (Figure 4a). This was associated with an increased APX activity mediating enzymatic ROS detoxification (44%, Figure 4c) but declining levels of total antioxidants (−59%, Figure 4b). Proline (Figure 4d) and glycine betaine (Figure 4e) as indicators for osmotic adjustment in drought-stressed plants increased by 319% and 179%, respectively. Geohumus application further increased APX activity and total antioxidants without an additional increase by co-inoculation with the microbial consortium (Figure 4b,c). The increased production of ROS-protective metabolites was associated with a significant decline in H_2_O_2_ concentrations in the leaf tissue (Figure 4a). Geohumus application also increased the proline concentrations (138%) and even reduced the concentrations of glycine betaine (−52%). Both metabolites were further increased by the application of the microbial inoculants (Figure 4d,e). Only the combined application of Geohumus and microbial inoculants resulted in the maximal accumulation of all investigated antioxidants and osmoprotectants.

Since plant drought adaptations are mediated by phytohormonal signaling, hormonal profiles of the leaf tissue were recorded. Two weeks after drought stress recovery, the concentrations of indole acetic (IAA), abscisic (ABA), jasmonic (JA), and salicylic acids (SA) were significantly increased compared to the well-watered plants, while gibberellic acid and zeatin remained below the detection limit. Geohumus application further increased the concentrations of IAA, ABA, JA, and SA without additional changes by inoculation with the microbial consortium (Figure 5).

#### 3.2.2. Sanoplant Experiment

Excess Mn concentrations in the leaf tissue and reduced tuber biomass after SC application recorded in the Geohumus experiment may indicate negative side effects of SC applications at higher soil moisture levels promoting soil Mn availability. Consequently, in the Sanoplant experiment, inoculant and SC effects were investigated for drought-stressed plants and additionally under well-watered conditions to assess potential limitations.

##### Plant Growth, AM Colonization, and Nutritional Status

Similar to the Geohumus experiment, drought stress tended to reduce tuber biomass by 56% and tuber size, while no differences were recorded for the number of tubers and stolons (Figure 6). The Sanoplant–inoculant combination significantly increased tuber biomass (82%) over the untreated control and tuber size even under well-watered conditions with a similar trend for drought-stressed plants (81%). Moreover, the number of stolons but not the number of tubers was significantly increased by Sanoplant application in the drought stress treatment (133%).

Drought stress reduced total plant biomass by 29%. Under well-watered conditions, the microbial consortium increased plant biomass by 30%. However, additional Sanoplant application had no further effects on plant biomass production, both in well-watered and drought-affected plants. Increased tuber biomass production in the Sanoplant treatments was associated with a concomitant reduction in shoot and root biomass, while total plant biomass remained unaffected (Table 5).

Drought stress increased total root length by 47% with a similar trend for root biomass. Under well-watered conditions, the microbial consortium significantly increased root biomass (71.5%) and root length (68%), while the combined application with Sanoplant had no significant effects on root growth. By contrast, the Sanoplant–inoculant combination stimulated AM establishment (colonization rate, external hyphae, vesicle formation), particularly in well-watered plants, with a similar trend under drought stress (Table 6).

Apart from N, mineral nutrients reached the sufficiency range in all investigated treatments (Table 7). The Sanoplant–inoculant combination significantly reduced shoot N concentrations and contents compared with the untreated controls, both in well-watered and drought-affected plants. In the face of the beneficial effects of the silicatic soil conditioner Sanoplant in this experiment and the well-documented drought-protective effects of Si in the literature [49], the silicone (Si) status of the plants status was also documented. However, no significant differences were recorded for Si, although plants treated with Sanoplant tended to show elevated Si concentrations (37–56%, Table 7). In contrast to the Geohumus experiment, no excessive accumulation of Mn and Fe was detectable in the soil conditioner treatments.

Cumulative determination of mineral nutrient accumulation in the tubers and shoot tissue was performed to assess the potential contribution of the AM inoculant to plant P acquisition. The Sanoplant–inoculant combination only marginally increased total P contents (2–13%) despite improved AM colonization but significantly reduced the K concentrations in the tuber tissue of drought-stressed plants (−15.4%).

##### Physiological Stress Indicators

Similar to the Geohumus experiment, leaf H_2_O_2_ concentrations (Figure 7a) were still massively increased after drought stress recovery in comparison with well-watered plants (684%). The combined application of microbial inoculants with Sanoplant significantly reduced the drought-induced H_2_O_2_ accumulation by 60% (Figure 7a), associated with increased enzymatic (APX, 114%) and non-enzymatic (total antioxidants, 254%) ROS detoxification (Figure 7b,c), as well as increased accumulation of proline (109%) and glycine betaine (125%) as indicators for osmotic adjustment (Figure 7d,e).

Moreover, the microbial inoculants exerted a priming effect even on well-watered plants, reflected by increased APX activity (89%), proline (53%), and glycine betaine accumulation (150%), which was not further increased by additional Sanoplant application (Figure 7a–c).

Concerning hormonal profiles, drought stress significantly increased the leaf concentrations of IAA, ABA, JA, and SA, which were further increased by the Sanoplant–inoculant combination. Zeatin and GA remained below the detection limit. However, under well-watered conditions neither the microbial inoculants nor their combination with Sanoplant affected the levels of the respective phytohormones (Figure 8). A similar pattern was recorded also for the root tissues (Appendix A).

## 4. Discussion

### 4.1. Impact of Soil Conditioners on Soil Water Relationships

Contrary to our working hypothesis and the various literature reports [50,51,52], the investigated soil conditioners (SC) were not able to increase the substrate WHC significantly, neither in the unplanted substrate nor in the rhizosphere soil at the final harvest (Table 1). This may be attributed to the soil properties since the benefits of SC applications concerning improved water retention have been preferentially observed in sandy substrates [40,53,54], while silty loam was used in our study. Moreover, the application of fertilizer salts and the presence of multivalent ions can negatively affect the WHC of SC products [40]. However, at least for the polyacrylate SC “Geohumus” (GH), the formation of local hotspots with high water content, represented by large SC-soil aggregates closely sticking to the roots (Appendix A), cannot be excluded. These aggregates could not be removed from the fine roots during collection of the rhizosphere soil and were consequently not considered for WHC determination. Nevertheless, they may represent hotspots with high substrate moisture and reduced O_2_ availability, particularly during the experimental periods with sufficient water supply. A related decline in redox potential may increase the highly plant-available reduced forms of micronutrients, such as Fe and Mn originating from the soil substrate and/or the volcanic minerals used in the Geohumus formulation. Consequently, excess shoot concentrations of Zn and Mn close to the toxicity level were recorded in the respective plants (Table 4). This was not the case in potato plants with “Sanoplant” (SP) supply (Table 7), which was not associated with the formation of sticky soil aggregates (Appendix A).

### 4.2. Oxidative Stress Defense and Hormonal Status

Despite the absence of beneficial SC effects on substrate WHC, the SC treatments provided other drought-protective functions, particularly in combination with the selected microbial inoculants *Rhizophagus irregularis* MUCL41833 + *Pseudomonas brassicacearum* 3Re2-7 (AM+3Re27). This was reflected in a reduction in irreversible drought-induced leaf damage caused by oxidative stress and increased shoot biomass (Figure 1). Accordingly, a common feature of the selected drought stress protectants was the induction of various physiological adaptations involved in ROS detoxification. This included increased ascorbate peroxidase (APX) activity, known as a key enzyme for H_2_O_2_ degradation under drought stress conditions [55], and increased production of antioxidants in the leaf tissue, consequently associated with a reduction in drought-induced ROS (H_2_O_2_) accumulation. Also, metabolites involved in osmotic adjustment, such as glycine betaine and proline [56], were significantly increased (Figure 4 and Figure 7). The protective effects were still detectable two weeks after recovery from drought stress treatments. APX activity, proline, and glycine betaine accumulation increased even in well-watered plants. This indicates a potential for the induction of longer-lasting systemic stress priming effects. Maximum protection against drought-induced leaf damage was achieved by combining the selected microbial inoculants with the soil conditioners (Figure 1 and Figure 4) suggesting complementary or synergistic functions of the applied agents.

Drought stress massively increased ROS (H_2_O_2_) accumulation, associated with a decline in total antioxidants and only a moderate increase in APX activity, proline, and glycine betaine accumulation (Figure 4 and Figure 7). The selected microbial inoculants partially compensated for the loss of antioxidants and further increased APX activity, proline, and glycine betaine accumulation. Similar protective effects induced by AM–*Pseudomonas* consortia have been previously reported for field-grown maize [57] or *Cupressus arizonica* [58]. Interestingly, the SC treatment with Geohumus also increased APX activity and accumulation of antioxidants and proline, but not of glycine betaine, associated with a moderate reduction of H_2_O_2_ levels (Figure 4). This suggests a stimulatory effect of Geohumus on ROS defense, potentially via the induction of latent Mn and Fe toxicity recorded in potato plants with Geohumus application, which was not detectable in Sanoplant treatments (Table 4 and Table 7).

This was associated with the induction of ROS detoxification and osmotic adjustment as drought-protective physiological responses. Tissue concentrations of hormones involved in stress signaling, and the induction of the respective responses, such as ABA, JA, and SA [59,60,61], were increased in drought-affected plants and particularly after application of the drought-protective agents (Figure 5 and Figure 8).

### 4.3. Biomass Partitioning, Tuber Formation, and Plant-Nutritional Status

According to the first view in the present study, increased shoot biomass and healthy leaf development suggest plant growth promotion by application of the selected drought protectants (Figure 1). However, a more comprehensive look at plant biomass production including shoot, tubers, and roots revealed that biomass allocation between different organs, not total plant biomass, was influenced by the experimental treatments, with selective effects depending on the applied SC type. Within a given stress level (drought-stressed or well-watered plants), total plant biomass was not significantly affected by SC treatments (Table 3 and Table 5). However, compared with untreated controls, the application of Geohumus decreased tuber biomass while shoot biomass increased (Table 3). By contrast, Sanoplant treatments increased tuber biomass (even under well-watered conditions) at the expense of shoot and root biomass production (Table 5, Figure 6).

The dominant effect of SC applications on biomass partitioning is further underlined by the fact that tuber numbers were not significantly changed (Table 2 and Figure 6), indicating that tuber initiation was obviously not affected. The negative effect of Geohumus on tuber growth may be explained by the excessive shoot concentrations of Mn and Fe, which were not detectable in the Sanoplant treatments (Table 4 and Table 7). Negative effects on tuber formation due to excess Fe [62] and Mn have been previously reported, e.g., under soil acidity [63]. The related activation of oxidative stress defense triggered by Geohumus application with similar responses additionally induced by the microbial inoculants (Figure 4) may increase the demand for photoassimilates in the affected shoot tissue and thereby limit carbon availability for tuber growth. Preferential resource allocation to the shoot tissue may be also reflected by a general trend for higher shoot contents of mineral nutrients in the Geohumus variants compared with the untreated controls (Appendix A). An opposite trend was recorded for the Sanoplant treatments, favoring resource allocation towards tuber growth (Appendix A).

Another exclusive feature of the Sanoplant variants was the promotion of AM colonization (Table 6 and Figure 2), which was not detectable in the Geohumus treatments (Figure 2 and Figure 3). The effects were apparent particularly in well-watered plants (Table 6), excluding improved soil water retention by the Sanoplant application as a potential cause. Consequently, the observed benefits of Sanoplant on AM colonization must be attributed to other, yet unknown alterations of soil properties. In this context, various studies reported increased colonization of the polymeric SC network with beneficial microorganisms [30,31,32] also including genera with members of mycorrhizal helper bacteria (MHB) with the ability to support AM colonization.

The beneficial effects of AM fungi and combinations with MHB on stress alleviation in potatoes are well documented in the literature reviewed by [64,65]. Major reported benefits comprise improved nutrient acquisition (particularly P and micronutrients) and water use efficiency, improved ROS detoxification, osmotic adjustment, and photosynthetic efficiency, disease suppression, and increased soil water retention in the rhizosphere.

However, in our study, the selective stimulation of tuber growth, associated with improved AM colonization by combining microbial inoculants with Sanoplant, was obviously not related to improvements in nutrient acquisition (Table 7). This applied also to the well-characterized beneficial element silicon (Si) with numerous stress-protective functions [49,66] as a major constituent of the selected SC products. Hence, the observed systemic stress priming effects of the inoculant–Sanoplant combination concerning ROS detoxification and osmotic adjustment (Figure 7), together with hormonal signatures (Figure 8) characteristic for both, induction of drought stress adaptations, and stimulation of tuberization [67,68], may be regarded as critical stress-protective factors.

Recently, it was demonstrated that downregulation of a sucrose exporter (SWEET11) promoting symplastic sucrose transport to developing tubers by inhibition of sucrose efflux into the apoplast, may act as a switch to mediate resource allocation for tuber development [69]. Interestingly, a downregulation of SWEET11 in potato has been similarly reported for inoculation with *Rhizophagus irregularis* [70] as AM fungus, used also in our study. This may explain the observed shift in biomass allocation towards tuber formation after promotion of AM colonization by Sanoplant application.

### 4.4. Conclusions

An increasing number of studies suggest that SC applications have the potential to enhance potato growth and tuber yield, especially in semiarid regions. Recently, Smagin et al. [53] reported that the application of a synthetic hydrogel soil conditioner significantly increased tuber yield (6–15 t ha^−1^), saved irrigation water up to two-fold, and protected plants against pathogens, particularly against late blight infestation. Moreover, the soil conditioner treatment potentially retained agrochemicals and minimized the leaching Overview representing the experimental setup of the studyloss of mineral fertilizers and pesticides. Similarly, the application of synthetic or natural soil conditioners significantly increased the soil enzymatic activity of phosphatase, catalase, urease, invertase, and soil microbial biomass of N, C, and P [71,72,73].

However, the data presented in this study demonstrate that the proposed benefits of SC application to increase the soil water-holding capacity under drought conditions cannot be generalized for all soil types. Nevertheless, depending on the selected SC type, beneficial effects on potato tuber yield can be achieved even under well-watered conditions, suggesting effects independent of soil–water relationships. Stimulation of AM colonization, stress priming effects concerning ROS detoxification and osmotic adjustment, and modifications of hormonal balances are among the potential factors promoting tuber growth and drought stress resilience. However, negative effects are also possible, as demonstrated by excessive Mn and Fe accumulation and inhibited tuber growth in the GH treatments. This underlines the importance of the selection and identification of compatible combinations of SCs, soil substrates, microbial inoculants, and host plants for the effective exploitation of synergisms as a target for future research.

## Figures and Tables

**Figure 1 microorganisms-12-02128-f001:**
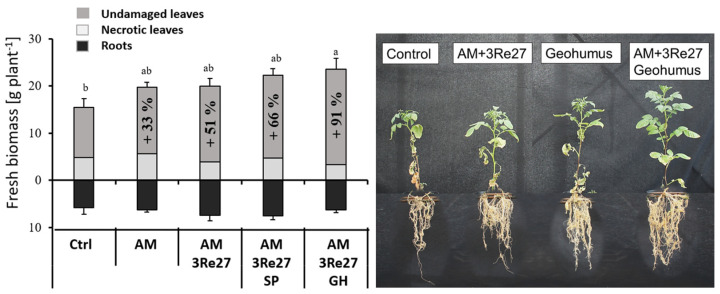
Plant habitus (**right**) and fresh plant biomass (**left**) with proportions (%) of undamaged leaves in potato plants after recovery from a 34 d drought stress period with and without application of microbial inoculants (*Rhizophagus irregularis* MUCL41833 = AM; *Pseudomonas brassicacearum* 3Re2-7 = 3Re27) and silicatic soil conditioners (Sanoplant = SP; Geohumus = GH). Means and SE of five replicates. Different lowercase characters above the bars indicate significant differences in shoot biomass (*p* ≤ 0.05).

**Figure 2 microorganisms-12-02128-f002:**
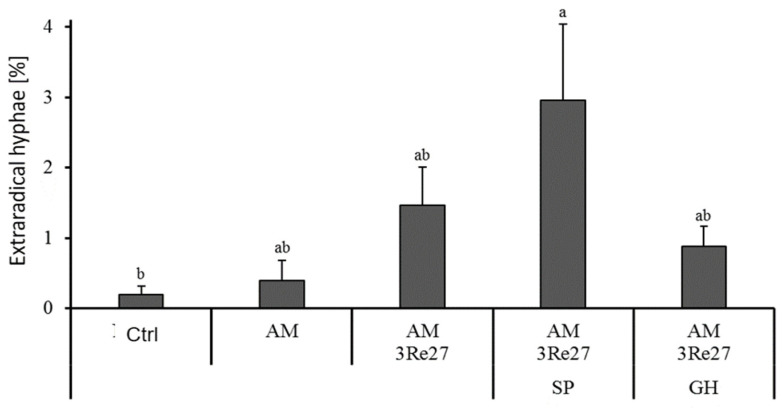
Extraradical AM hyphae (*Rhizophagus irregularis* MUCL41833 = AM; *Pseudomonas brassicacearum* 3Re2-7 = 3Re27) and silicatic soil conditioners (Sanoplant = SP; Geohumus = GH). Different lowercase characters above the bars indicate significant differences (*p* ≤ 0.05).

**Figure 3 microorganisms-12-02128-f003:**
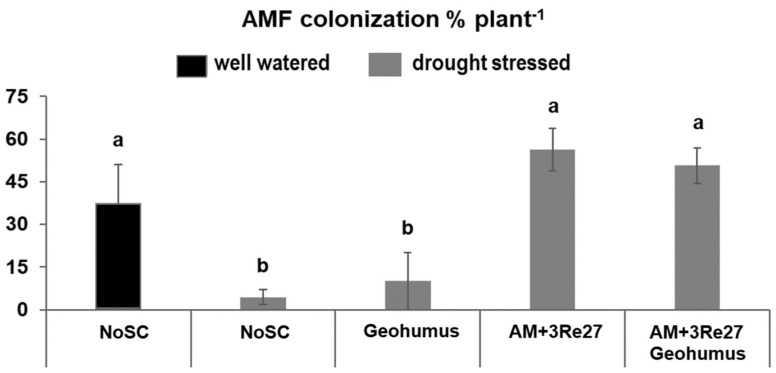
AM root colonization of potato plants exposed to drought stress with and without (AM = *Rhizophagus irregularis* MUCL41833, 3Re27 = *Pseudomonas brassicacearum* 3Re2-7, and silicatic soil conditioner Geohumus and NoSC = without soil conditioner). Different lowercase letters above the bars indicate significant differences between means (*p* ≤ 0.05).

**Figure 4 microorganisms-12-02128-f004:**
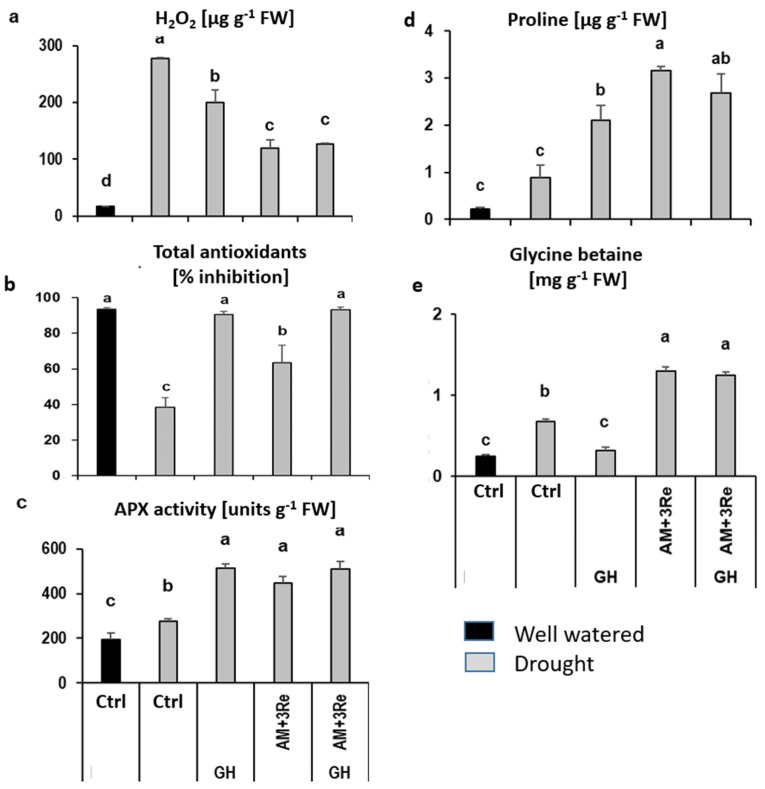
Effect of microbial inoculants and Geohumus (**a**) H_2_O_2_, (**b**) total antioxidants, (**c**) APX activity, (**d**) proline, and (**e**) glycine betaine accumulation of potato leaves two weeks after drought stress recovery (AM = *R. irregularis* MUCL 41833, 3Re: = P. *brassicacearum* 3Re2-7, and GH: Geohumus). Different lowercase letters above the bars indicate significant differences between means (*p* ≤ 0.05).

**Figure 5 microorganisms-12-02128-f005:**
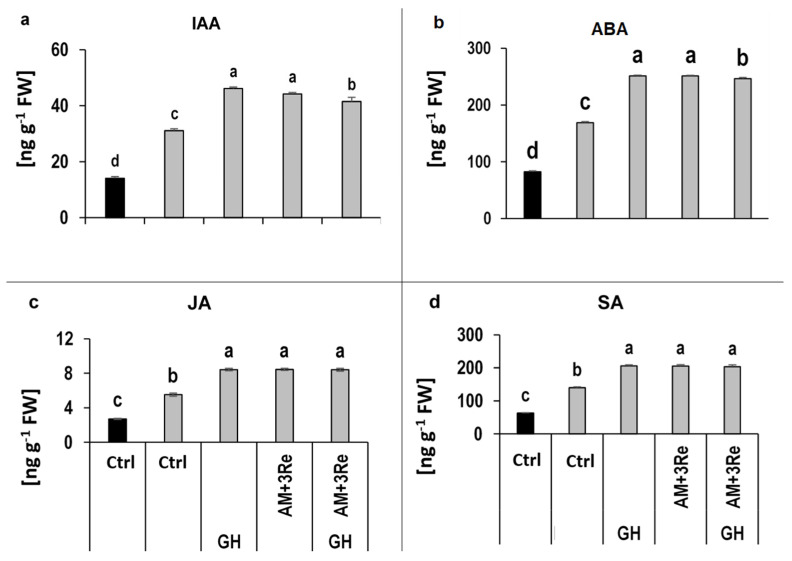
Effect of microbial inoculants and Geohumus on (**a**) IAA, (**b**) ABA, (**c**) JA, and (**d**) SA concentrations in leaf tissues of potato under well-watered conditions (black bars) and two weeks after drought stress recovery (gray bars). AM = *R. irregularis* MUCL 41833, 3Re: *P. brassicacearum* = 3Re2-7, and GH: Geohumus. Different lowercase letters above the bars indicate significant differences between means (*p* ≤ 0.05).

**Figure 6 microorganisms-12-02128-f006:**
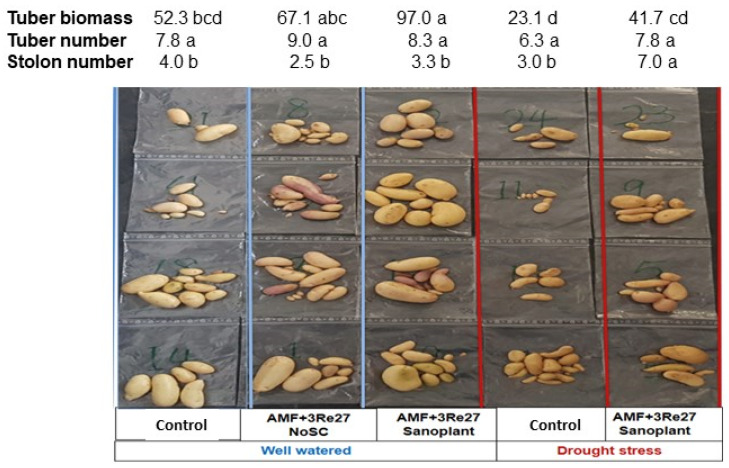
Effect of microbial inoculants and Sanoplant on potato fresh tuber biomass (g plant^−1^) and number of tubers and stolons per plant under well-watered and drought stress conditions (AMF = *R. irregularis* MUCL 41833, 3Re27 = *P. brassicacearum* 3Re2-7, and soil conditioner Sanoplant). Different characters within the same row indicate significant differences between means (*p* ≤ 0.05).

**Figure 7 microorganisms-12-02128-f007:**
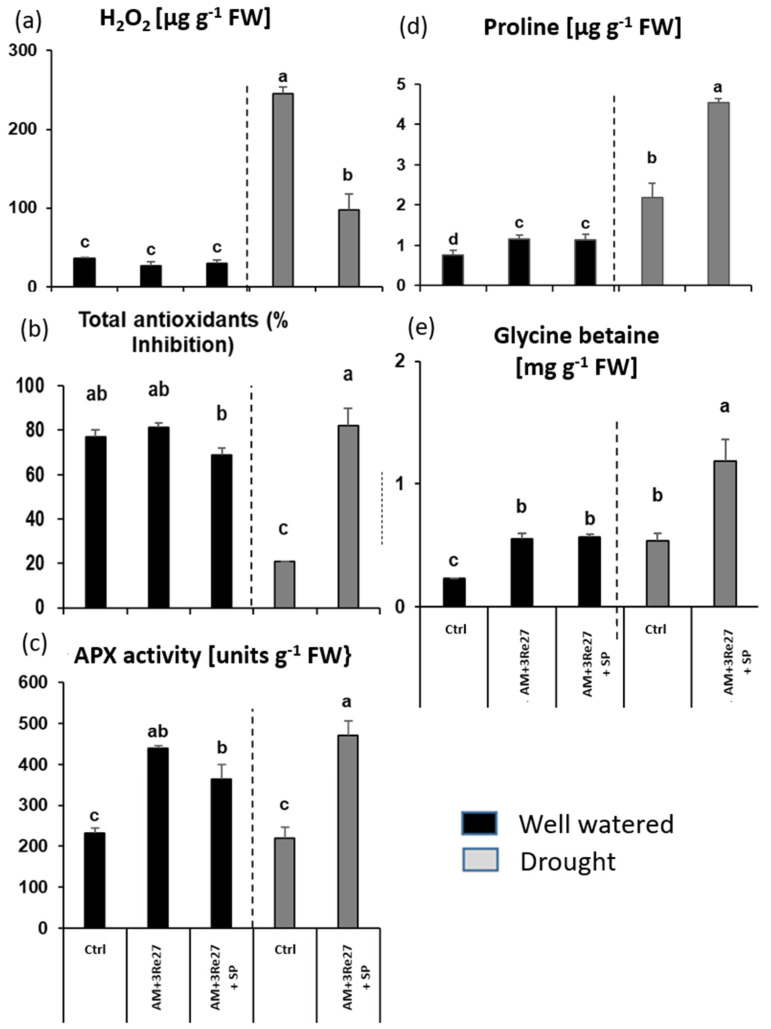
Effect of microbial inoculants and Sanoplant on leaves. (**a**) H_2_O_2_, (**b**) total antioxidants, (**c**) APX activity, (**d**) proline, and (**e**) glycine betaine of potato (AM = *R. irregularis* MUCL 41833, 3Re27 = *P. brassicacearum* 3Re2-7, SP = Sanoplant). Different lowercase letters above the bars indicate significant differences between means (*p* ≤ 0.05).

**Figure 8 microorganisms-12-02128-f008:**
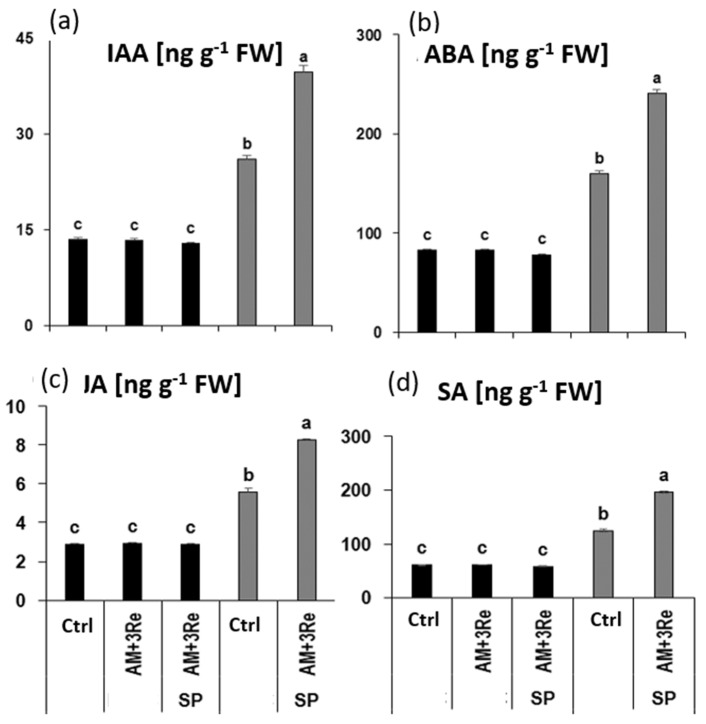
Effect of microbial inoculants and Geohumus on (**a**) IAA, (**b**) ABA, (**c**) JA, and (**d**) SA concentrations in leaf tissues of potato under well-watered conditions (black bars) and two weeks after drought stress recovery (gray bars). AM = *Rhizophagus irregularis* MUCL 41833, 3Re27 = *P. brassicacearum* 3Re2-7, SP = Sanoplant). Different lowercase letters above the bars indicate significant differences between means (*p* ≤ 0.05).

**Table 1 microorganisms-12-02128-t001:** Gravimetric soil water content [g H_2_O 100 g dry soil^−1^] at different levels of substrate water-holding capacity (WHC) of unplanted substrate (A) and rhizosphere soil with and without application of soil conditioners (SCs), Sanoplsant (SP), or Geohumus (GH), determined after [36,39]. n.s = not signifivant.

**(A) SC Treatment**	**100% WHC**	**60% WHC**	**20% WHC**
Control	35.2 n.s.	21.1 n.s.	7.0 n.s
SP	37.5 n.s	22.5 n.s	7.5 n.s
GH	37.5 n.s.	22.5 n.s	7.5 n.s
**(B) SC Treatment**	**100% WHC**	**60% WHC**	**20% WHC**
Control	39.3 n.s	23.6 n.s	7.9 n.s
SP	39.9 n.s	23.9 n.s	8.0 n.s
AM+3Re27+SP	38.7 n.s	23.2 n.s	7.7 n.s
GH	38.9 n.s	23.3 n.s	7.8 n.s
AM+3Re27+GH	38.7 n.s	23.2 n.s	7.7 n.s

**Table 2 microorganisms-12-02128-t002:** Total fresh tuber biomass and tuber numbers of potato plants with microbial inoculants and the soil conditioner Geohumus (GH); AM = *Rhizophagus irregularis* MUCL41833; and 3Re2-7 = *Pseudomonas brassicacearum* 3Re2-7 grown under well-watered and drought-stressed conditions. Means ± SE of four replicates. Different lowercase letters within the same column indicate significant differences between means (*p* ≤ 0.05).

Treatments	Tuber Biomass [g Plant^−1^]	Tuber Number [Plant^−1^]
Well watered	216.2 ± 60.1 a	25.5 ± 13.5 a
Drought	121.6 ± 34.1 ab	15.8 ± 4.3 a
Drought + GH	79.9 ± 23.5 b	10.5 ± 3.5 a
Drought + AM + 3Re2-7	213.8 ± 30.3 a	9.5 ± 1.6 a
Drought + AM + 3Re2-7 + GH	67.5 ± 24.4 b	7.8 ± 0.9 a

**Table 3 microorganisms-12-02128-t003:** Total fresh plant biomass and relative biomass distribution of potato plants with microbial inoculants (AM = *Rhizophagus. irregularis* MUCL 41833, 3Re27 = *Pseudomonas. brassicacearum* 3Re2-7) and silicatic soil conditioner (GH = Geohumus) under WW = well-watered and drought stress conditions. Different lowercase letters within the row indicate significant differences between means (*p* ≤ 0.05).

	WW	Drought	GH Drought	AM + 3Re2-7 Drought	AM +3 Re2-7 +GH Drought
Total FW [g plant^−1^]	490.8 a	388.0 ab	368.9 b	522.0 a	481.2 ab
Relative biomass distribution [%]					
Shoot	49.1	50.8	61.3	48.3	70.4
Tubers	44.1	31.3	21.6	41.0	14.0
Roots	6.8	17.7	17.1	10.7	9.2

**Table 4 microorganisms-12-02128-t004:** Shoot nutrient concentrations of potato plants grown under well-watered and drought stress conditions. GH: Geohumus, AM: *R. irregularis* MUCL 41833, and 3Re27: *P. brassicacearum* 3Re2-7, deficiency thresholds according to [48]. Different lowercase letters within the same column indicate significant differences between means (*p* ≤ 0.05).

Treatments	P	K	N	Mg	Mn	Zn	Cu
Deficiency Threshold	2.5 mg·g^−1^	60 mg·g^−1^	30 mg·g^−1^	7.0 mg·g^−1^	30 mg·kg^−1^	30 mg·kg^−1^	3 mg·kg^−1^
Well watered	1.74 ± 0.18 a	30.0 ± 3.0 c	24.2 ± 4.1 c	6.5 ± 0.4 ab	27.96 ± 3.3 b	55.8 ± 5.3 b	14.8 ± 2.7 a
Drought	1.86 ± 0.10 a	37.3 ± 1.1 b	35.5 ± 1.8 b	6.6 ± 0.5 ab	45.08 ± 2.6 b	57.8 ± 5.1 b	14.2 ± 0.4 a
Drought + GH	2.00 ± 0.28 a	49.3 ± 3.1 a	43.2 ± 2.5 a	7.4 ± 0.8 a	419.42 ± 90.4 a	86.8 ± 7.0 a	16.1 ± 1.8 a
Drought + AM + 3Re2-7	1.66 ± 0.14 a	32.5 ± 1.8 bc	27.3 ± 1.0 c	5.1 ± 0.4 b	42.3 ± 8.3 b	51.9 ± 2.6 b	13.2 ± 0.5 a
Drought + AM + 3Re2-7+GH	2.14 ± 0.14 a	52.5 ± 2.0 a	41.9 ± 1.6 ab	6.4 ± 0.3 ab	413.27 ± 50.1 a	92.7 ± 11.4 a	16.8 ± 1.5 a

**Table 5 microorganisms-12-02128-t005:** Total fresh plant biomass and relative biomass distribution of potato plants exposed to drought. [AM: *Rhizophagus. irregularis* MUCL 41833; 3Re27: *Pseudomonas. brassicacearum* 3Re2-7; WW: well watered; SP: Sanoplant]. Different lowercase letters within the row indicate significant differences between means (*p* ≤ 0.05).

	WW Control	WW AM+3Re27	WW SP+AM+3Re27	Drought Control	AM+3Re27+SP Drought
Total FW [g]	187.1 bc	244.6a	212.9 ab	132.3 c	132.1 c
Relative biomass distribution [%]					
Shoot	62.8	60.5	45.0	63.9	56.8
Tubers	28.0	25.2	45.6	17.5	31.6
Roots	9.2	14.3	9.4	18.6	11.5

**Table 6 microorganisms-12-02128-t006:** Attributes of root morphology and AMF colonization [AM: *Rhizophagus. irregularis* MUCL 41833, 3Re27: *Pseudomonas. brassicacearum* 3Re2-7, SP: Sanoplant]. Different lowercase letters within the same column indicate significant differences between means (*p* ≤ 0.05).

Irrigation	Variants	Root Biomass [g plant^−1^]	Root Length [m plant^−1^]	AM Colonization [%]	Extraradical Hyphae [%]	Vesicles [%]
Well watered	Control	17.2 bc	51.9 d	16.7 c	14.5 b	8.5 b
	AM+3Re27	29.5 a	87.2 a	27.3 ab	17.5 ab	12.5 ab
	AM+3Re27+SP	20.1 abc	59.6 cd	33.8 a	28.5 a	18.5 a
Drought	Control	24.6 abc	76.3 abc	13.7 c	11.5 b	5.5 b
	AM+3Re27+SP	15.3 c	64.0 bcd	17.0 c	14.0 b	10.0 b

**Table 7 microorganisms-12-02128-t007:** Effect of microbial inoculants and soil conditioner on the mineral nutritional status in potato shoot tissue (dry matter) under well-watered and drought stress conditions [AM: *R. irregularis* MUCL 41833 and 3Re27: *P. brassicacearum* 3Re2-7, SP: Sanoplant]. Different lowercase letters within the same column indicate significant differences between means (*p* ≤ 0.05).

Treatments		P	K	N	Mg	Ca	Si	Mn	Zn
Deficiency Thresholds		2.5 mg·g^−1^	60 mg·g^−1^	39 mg·g^−1^	7.0 mg·g^−1^	6 mg·g^−1^	30 mg·g^−1^	30 μg g^−1^	30 μg g^−1^
**Well watered**	Control	2.81 ab	45.8 ab	20.6 bc	3.4 ab	11.1 b	0.49 a	15.0 c	50.5 bc
	AM+3Re27	2.75 b	45.8 ab	17.8 c	3.1 abc	12.8 ab	0.46 a	14.0 cd	44.4 c
	AM+3Re27+SP	2.76 b	45.0 b	13.7 d	2.6 c	14.7 ab	0.63 a	10.0 d	52.9 abc
**Drought**	Control	3.36 a	50.0 ab	34.2 a	3.6 a	16.2 a	0.58 a	38.2 a	56.2 ab
	AM+3Re27+SP	3.12 ab	54.3 a	22.0 b	2.8 bc	14.3 ab	0.92 a	30.9 b	61.4 a

## Data Availability

The original contributions presented in the study are included in this article/Appendix A, and further inquiries can be directed to the corresponding author.

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
