# Peer review of "Combination of Silicate-Based Soil Conditioners with Plant Growth-Promoting Microorganisms to Improve Drought Stress Resilience in Potato"

_microorganisms, 2024, doi:10.3390/microorganisms12112128_

Round 1
Reviewer 1 Report
Comments and Suggestions for Authors
The manuscript, entitled "Combination of silicatic soil conditioners with plant growth- promoting microorganisms to improve drought-stress resilience in potato" is a high-quality presentation of a series of scientific studies compiled with sufficient thoroughness. In the experiments, a potato test plant was used, as well as soil conditioning preparations (lignocellulose polysaccharides-SP) or (polyacrylate-GH) and PGPM microbes.
The introductory part and the part presenting the materials and methods used are sufficiently detailed.
The presentation of the results is sufficiently detailed and clearly illustrates the results achieved. Published test data also support the fact that the proposed benefits of silicate soil improvers for increasing soil WHC under drought conditions cannot be generalized to all soil types. Stimulation of AM colonization, stress-priming effects involving ROS detoxification and osmotic adjustment, and modification of hormonal balance are potential factors promoting tuber growth and drought stress resistance. However, negative effects are also possible (as evidenced by excessive accumulation of Mn and Fe) and inhibition of tuber growth during GH treatments. This emphasizes the importance of selecting and identifying compatible combinations of SCs, soil substrates, microbial inoculants and host plants to effectively exploit synergies.
In the Discussion chapter, I recommend a comparison with the results of previous, similar research on potatoes. Addendum related to this. Otherwise, the manuscript contains a sufficient number (61) of other references.
After the minor addition suggested above, I recommend publishing the manuscript in the form of a scientific article.
Reviewer 2 Report
Comments and Suggestions for Authors
Review on “Combination of silicatic soil conditioners with plant growth-promoting microorganisms to improve drought-stress resilience in potato” for manuscript ID microorganisms-3227829
In brief introduction the authors emphasize that potatoes are particularly sensitive to drought due to their shallow root systems, making them vulnerable to limited water availability. The study aims to investigate the effects of soil conditioners (two different types), which are hypothesized to improve soil water retention and support beneficial interactions between plants and inoculants (fungi and bacteria) under water deficit conditions.
Bulk citations aren’t good practice (L91), please select the most suitable references supporting your point. L102: please specify the particular species used in the present study. While the PGPR application is a common practice to help plant to adapt to various environment conditions, it’s better to narrow the literature review to drought-related studies. A brief summary of the previous study [30] would be helpful in the Intro.
Unfortunately, some recent studies have been missed. The following papers could help to improve the Intro:
· Liu J, Zhang J, Zhu M, Wan H, Chen Z, Yang N, Duan J, Wei Z, Hu T, Liu F. Effects of Plant Growth Promoting Rhizobacteria (PGPR) Strain Bacillus licheniformis with Biochar Amendment on Potato Growth and Water Use Efficiency under Reduced Irrigation Regime. Agronomy. 2022; 12(5):1031. https://doi.org/10.3390/agronomy12051031
· Waqar, A., Bano, A., & Ajmal, M. (2022). Effects of PGPR Bioinoculants, Hydrogel and Biochar on Growth and Physiology of Soybean under Drought Stress. Communications in Soil Science and Plant Analysis, 53(7), 826–847. https://doi.org/10.1080/00103624.2022.2028818
Major points:
Does it make sense to use Sanoplant and Geohumus together? According to the experimental plan, it seems to be a logical final step.
L103: biosafety and environmental impact of silicatic soil conditioners have to be covered also.
L294: how the colonization level was evaluated?
The scheme of three pot experiments would be helpful for the reader.
Minor points:
A “silicatic” term is rare in scientific literature, I suggest to replace it to more common variant.
At Figure 1 the root biomass showed in negative direction, at first sight it raises the question. The figure legend talks about “proportion”, but units show the absolute values (grams per plant).
Figure 5B has no units (guess the “%” should be there)
L259-262: font is inconsistent
Reviewer 3 Report
Comments and Suggestions for Authors
The manuscript is well written, and I did not have any trouble understanding it. The methods are sound, but I suggest some minor clarification as to the experimental procedure since it is somewhat hard to read, particularly as to the differences between the second and third experiments, which seem to be only which of the additives was used, but depend on the reader recalling inoculants names, which could be simplified if they are, as I understand it, the same strains in the three experiments. I strongly suggest using dry mass instead of fresh mess, to reduce data variability.
I also question the use of t-grouping for means comparison and not Tukey´s test since the t-test only compares two treatments.
I could not find the methods described for some results, such as those from Table 1, which is unacceptable. The table also includes a variable THK that is not accounted for.
The results should be similarly presented for both experiments. For example, while tuber biomass and numbers are given as bar charts for the geohumus experiment, they are presented as numbers and photos for the sanoplant experiment, making a visual comparison between the experiments nearly impossible for the reader.
Variables are also presented for one experiment and not for the other, such as biomass distribution, which I also do not recall being discussed in the methods.
The discussion largely repeats and compares the results with the literature, particularly in its first section.
Reviewer 4 Report
Comments and Suggestions for Authors
Submission ID: microorganisms-3227829
Title of the manuscript: "Combination of silicatic soil conditioners with plant growth promoting microorganisms to improve drought-stress resilience in potato".
This study evaluated the impact of two commercial silicatic soil conditioners and PGPM inoculants on potato plants under drought stress. The subject is very interesting, and the manuscript is well-written and full of work. I highly recommended publishing this manuscript in “Microorganisms”. However, there are still some issues that need to be addressed, I suggest major revision.
Specific comments
L43: Please arrange the keywords in alphabetical order.
L43: Sometimes authors use PGPMs and in other part use PGPR, please use one form in the whole manuscript.
L43: Please define acronym PGPR, plant growth-promoting microorganisms (PGPR).
L46-112: In the introduction, I suggest the author to start by the potato plant and its economic benefits, then extend to its problem with drought and other abiotic stresses in general, then you can give the solution for the problem using silicatic soil conditioners and PGPMs application, ending by solid hypothesis to give a brief about the novelty of the study.
L54: Not clear. Please extend here the role of glamalin.
L104-112: The authors should refer to the mechanistic used to achieve their goal, then may refer some lack in the previous study regarding some aspects (Novelty).
L115: Which years??
L181-183: How many mycorrhizal spores per gram??? Please clarify?
L250: Please follow the journal notation in citing the references.
L259-262: Unify font size.
L267: P should be italic in the whole manuscript.
L303: How did the authors determine extraradical AM hyphae%? Please include these details in M&M.
Table3: Please add this sentence in the table caption “Different letters within the same column indicate significant differences between means.
Figure 5: Please try to enhance the quality of the figure – The unit of total antioxidants is absent – I suggest transferring the figure name such as H2O2, proline to the Y axis before the units.
I appreciate the well-written discussion.
L607-618: I suggest moving this part in a separate section under the title “conclusion”.
Regards.
Comments on the Quality of English LanguageMinor editing of English language required
Round 2
Reviewer 2 Report
Comments and Suggestions for Authors
I would like to thank the authors for the improving the manuscript, but some concerns remain to be addressed.
Suppl. Table S3: "Mn cone" to "Mn conc"
The term "silicatic" remains in Suppl. Table S4, S5
L580: missing space; L591: "H2O2" subscript required
L601: "synergistic functions of the amendments" - please rephrase for clarity; If the term "amendment" fits here?
L681: "suggest the potential", the potential cannot be suggested
L682: Smagin et. al is the [53] ref.
Author Response
For research article
|
Response to Reviewer 2 Comments
|
||
|
1. Summary |
|
|
|
Thank you very much for taking the time to review this manuscript. Please find the detailed responses below and the corresponding revisions/corrections highlighted/in track changes in the re-submitted files. [This is only a recommended summary. Please feel free to adjust it. We do suggest maintaining a neutral tone and thanking the reviewers for their contribution although the comments may be negative or off-target. If you disagree with the reviewer's comments please include any concerns you may have in the letter to the Academic Editor.]
|
||
|
2. Questions for General Evaluation |
Reviewer’s Evaluation |
Response and Revisions |
|
Does the introduction provide sufficient background and include all relevant references? |
Yes/Can be improved/Must be improved/Not applicable |
[Please give your response if necessary. Or you can also give your corresponding response in the point-by-point response letter. The same as below] |
|
Are all the cited references relevant to the research? |
Yes/Can be improved/Must be improved/Not applicable |
|
|
Is the research design appropriate? |
Yes/Can be improved/Must be improved/Not applicable |
|
|
Are the methods adequately described? |
Yes/Can be improved/Must be improved/Not applicable |
|
|
Are the results clearly presented? |
Yes/Can be improved/Must be improved/Not applicable |
|
|
Are the conclusions supported by the results? |
Yes/Can be improved/Must be improved/Not applicable |
|
|
3. Point-by-point response to Comments and Suggestions for Authors |
||
|
Comments 1: Suppl. Table S3: "Mn cone" to "Mn conc" The term "silicatic" remains in Suppl. Table S4, S5
Response 1: Suggested corrections in the Supplementary data have been incorporated
Comments 2: L580: missing space; L591: "H2O2" subscript required Response 2: corrected
Comments 3: L601: "synergistic functions of the amendments" - please rephrase for clarity; If the term "amendment" fits here? Response 3: rephrased to "synergistic functions of the applied agents"
Comments 4: L681: "suggest the potential", the potential cannot be suggested Response 4: rephrased to: "An increasing number of studies suggest that SC applications have the potential to enhance potato growth and tuber yield, especially in semi-arid regions."
Comments 5: L682: Smagin et. al is the [53] ref. Response 5: corrected
|
||
Reviewer 4 Report
Comments and Suggestions for Authors
This is the second time I have evaluated this manuscript. The authors addressed all my comments, and the manuscript has been noticeably improved. Many thanks for their contribution.
Author Response
For research article
|
Response to Reviewer 4 Comments
|
||
|
1. Summary |
|
|
|
Thank you very much for taking the time to review this manuscript. Please find the detailed responses below and the corresponding revisions/corrections highlighted/in track changes in the re-submitted files.
|
||
|
2. Questions for General Evaluation |
Reviewer’s Evaluation |
Response and Revisions |
|
Does the introduction provide sufficient background and include all relevant references? |
Yes/Can be improved/Must be improved/Not applicable |
[Please give your response if necessary. Or you can also give your corresponding response in the point-by-point response letter. The same as below] |
|
Are all the cited references relevant to the research? |
Yes/Can be improved/Must be improved/Not applicable |
|
|
Is the research design appropriate? |
Yes/Can be improved/Must be improved/Not applicable |
|
|
Are the methods adequately described? |
Yes/Can be improved/Must be improved/Not applicable |
|
|
Are the results clearly presented? |
Yes/Can be improved/Must be improved/Not applicable |
|
|
Are the conclusions supported by the results? |
Yes/Can be improved/Must be improved/Not applicable |
|
|
3. Point-by-point response to Comments and Suggestions for Authors |
||
|
Comments 1: No further modifications requested by reviewer 4
|
||